# A Genetic and Environmental Analysis of Inflammatory Factors in Chronic Widespread Pain Using the TwinsUK Cohort

**DOI:** 10.3390/biom15020155

**Published:** 2025-01-21

**Authors:** Stacey S. Cherny, Gregory Livshits, Frances M. K. Williams

**Affiliations:** 1Department of Anatomy and Anthropology, Tel Aviv University, Tel Aviv 6997801, Israel; cherny@tauex.tau.ac.il; 2Department of Morphological Sciences, Adelson School of Medicine, Ariel University, Ariel 4077625, Israel; zviliv@ariel.ac.il; 3Department of Twin Research and Genetic Epidemiology, King’s College London, London SE1 7EH, UK

**Keywords:** chronic widespread pain, CWP, fibromyalgia, Olink, inflammation, TwinsUK

## Abstract

Chronic widespread musculoskeletal pain (CWP), a significant health issue affecting individuals and society, is often diagnosed as part of fibromyalgia but is not generally considered inflammatory. This study investigated the relationship between blood-based inflammatory factors and CWP in 904 individuals from the TwinsUK cohort. Participants, free of major inflammatory conditions, completed questionnaires to assess CWP. Plasma samples were analysed using the Olink panel, alongside assays for C-reactive protein (CRP) and Apolipoproteins A1 and B. No significant associations were observed between CWP and inflammatory factors after adjusting for multiple testing. Twin modelling revealed significant heritability for both CWP and inflammatory factors, with genetic covariance observed between CWP and several inflammatory factors. Additive Bayesian network modelling suggested that any association between CWP and inflammatory factors is mediated by body mass index (BMI). These findings emphasize the complexity of CWP and its potential reliance on factors beyond inflammation, such as BMI, which strongly correlates with CRP and other inflammatory markers. Future research should explore additional molecular, genetic, and environmental contributors to CWP variability and investigate clinical factors or covariates that may obscure relationships with inflammation, providing a more comprehensive understanding of this multifaceted condition.

## 1. Introduction

Chronic widespread musculoskeletal pain (CWP), for which no apparent cause can be identified [1], is a pervasive and enigmatic health challenge, exerting a profound impact on individuals and society as a whole. Meta-analyses have estimated the population prevalence of CWP to be around 10%, with prevalence somewhat higher in women than men and increasing with age [2,3]. Chronic pain has been found to be an enormous economic burden to the healthcare system, increasing per capita expenditures on healthcare in the USA by nearly USD 300 [4]. CWP has an important impact on the quality of life of vast numbers of people [5]. The complexities of CWP go beyond the mere experience of pain, encompassing a spectrum of physical, psychological, and societal consequences. Understanding the aetiology and risk factors for CWP is essential for developing targeted interventions that address the root causes of this complex condition, offering hope for improved quality of life for those affected.

We have previously shown the existence of a significant association between CWP and steroid hormone levels [6]. Our study also suggested that steroid hormone abnormalities likely result from pain rather than causing it and that epiandrosterone sulphate can probably be used as a biomarker of the condition. Furthermore, our study of DNA methylation in CWP suggested the involvement of neurological pathways in this condition [7]. On the other hand, androgens and oestrogens have been shown to exert significant immunomodulatory and anti-allergy effects [8,9] and may account in part for the raised prevalence of CWP in women. In addition, our previous study of UK twins demonstrated that CWP cases exhibited greater adiposity than controls [6], and elevated body mass index (BMI) has been found to be a risk factor for CWP in a longitudinal study [10].

The link between pain and inflammation is well acknowledged, with multiple studies having illuminated the involvement of pro-inflammatory cytokines in the pathophysiology of pain [11,12]. Much less is known about the relationship between CWP and inflammation, and even less still involving systematic study of various inflammatory factors in CWP patients. Two reviews of the literature on the relationship between cytokines and fibromyalgia, of which CWP is a feature, have been performed, with one concluding that some inflammatory factors are elevated in those with fibromyalgia as compared to controls [13] and the other unable to make any definitive conclusions [14]. However, the most recent meta-analysis of fibromyalgia examined nine biomarkers and found some evidence of differences between cases and controls for seven of them, including IL-10, IL-6, IL-8, TNF-alpha, IFN.gamma, and CRP, which are associated with the inflammatory process [15]. This triggered our interest in investigating whether and to what extent inflammatory factors are associated with CWP in our community-based sample of middle-aged females, TwinsUK [16].

The Olink 96 inflammation panel allows for a hypothesis-free, high-throughput examination of many inflammatory factor levels in an individual. This panel has been used in the TwinsUK project for the study of various facets of health, including tinnitus [17] and, particularly of relevance to the present study, body mass index (BMI) [18]. While no statistically significant association was found for tinnitus, after stringent multiple testing correction, some interesting findings were found for BMI when it was analysed in conjunction with other omics data.

The aim of the present study is to examine the association between inflammation and CWP, using the largest sample to date, TwinsUK, assessed on the Olink 96 panel, along with measures of C-reactive protein (CRP) and Apolipoprotein A1 (ApoA1) and B (ApoB), and CWP, which has a relatively high prevalence (16.5%) in this all-female sample. By examining the molecular signatures associated with inflammation, we endeavoured to shed light on potential biomarkers that may serve as diagnostic indicators, prognostic tools, or therapeutic targets for CWP.

## 2. Materials and Methods

### 2.1. Sample

Participants were individuals from the UK Adult Twin Registry (TwinsUK) [16]. Founded in 1992, the TwinsUK registry comprises adult twin volunteers from the general population recruited through national media campaigns. The cohort is predominantly female (83%), middle-aged, mainly of Northern European descent, and comprises similar numbers of monozygotic and dizygotic same-sex twin pairs. Participants have been characterized for a variety of clinical and behavioural traits through clinical visits and annual health-based questionnaires collected by post and, more recently, electronically, including BMI. Twins from this registry have been shown to be similar to age-matched singletons for a range of health and lifestyle factors [19]. Twins were not specifically recruited for the current study of CWP; they were selected based on the availability of questionnaire assessments of CWP as well as having been assessed for inflammatory factors. All subjects provided written informed consent in accordance with the St Thomas’ Hospital Research Ethics Committee and were unaware of the precise hypotheses being tested.

### 2.2. Inflammatory Factors

Analysis of 96 proteins comprising the Olink Inflammatory (v.3021, Olink Proteomics, Uppsala, Sweden) panels was performed using PEA (proximity extension assay) technology on plasma samples. The PEA technique allows simultaneous assessment of proteins using oligonucleotide-labelled antibody probe pairs that bind to each protein within the sample, while demonstrating both high sensitivity and specificity [20,21]. The Olink panel compared favourably to two other multiplexed protein detection platforms, with both advantages and disadvantages [22]. The data were subjected to standard quality control (QC) procedures, resulting in 2346 female samples for 70 proteins in the inflammation panel available for analysis, with 26 proteins failing to reach the QC threshold.

For the sake of analysis, the proteins available for the present study were divided into four functional groups, namely, pro-inflammatory and anti-inflammatory mediators of inflammation, immunoregulatory factors, and multifunctional factors: (1) **Pro-inflammatory molecules** are essential for initiating the immune response to infectious agents. They stimulate immune cell recruitment and activation, increase vascular permeability, and induce fever, which are critical for fighting invading pathogens [23]. (2) **Anti-inflammatory molecules** are important in limiting inflammation and preventing tissue damage. They inhibit the production of pro-inflammatory cytokines and promote differentiation and activation of regulatory immune cells, which can help control the immune response and promote tissue repair [24]. (3) **Immunoregulatory molecules** possess both pro- and anti-inflammatory activities depending on the inflammation trigger, affected tissue, stage of inflammation, concomitant diseases, etc. [25]. (4) **Multifactorial molecules** are defined as those performing various functions, including participation in the inflammatory process.

In addition, data on CRP and two apolipoproteins, ApoA and ApoB, known for their roles in inflammation, were available for our study. CRP is one of the most widely recognized and frequently measured pro-inflammatory factors. ApoA was selected due to numerous findings demonstrating its anti-inflammatory effects both directly (against invading pathogenic microorganisms or their products) and indirectly (by modulation of functions of various immune cells, such as macrophages, neutrophils, and T lymphocytes) [26]. While the inflammatory role of ApoB remains unclear, it is often considered to have anti-inflammatory properties [27]. Table 1 provides a detailed list of factors, grouped by functional category.

### 2.3. Chronic Widespread Pain

The London Fibromyalgia Epidemiology Study Screening Questionnaire (LFESSQ) [28] was sent to twins for self-completion, without reference to the co-twin. This measure of CWP has been used in previous reports from TwinsUK [6,29]. A total of 1193 females (no males were assessed) completed the CWP assessment and provided Olink data that passed quality control (QC). Among these, 230 individuals were diagnosed with CWP, while 963 were CWP-negative. To refine the dataset, we included only cases where plasma samples were obtained within four years before or after the CWP diagnosis, resulting in 1019 individuals, 188 of whom were affected by CWP. Sample sizes for some inflammatory measures were smaller due to missing data. We further excluded individuals diagnosed with any of the following inflammatory conditions: rheumatoid arthritis, systemic lupus erythematosus, Crohn’s disease, and ulcerative colitis, reducing the sample to 904 women, of whom 149 had CWP (mean age 60.2, range 42.3–86.0) and 755 were unaffected with CWP (mean age 59.3, range 26.7–92.0).

### 2.4. Design of the Study

This study involves four stages of analysis to explore the relationship between CWP and a broad array of inflammatory factors. In the first stage, generalized linear mixed models were used to assess whether each inflammatory factor in the dataset predicts CWP. These analyses were conducted in two ways: first, by considering the entire set of inflammatory factors grouped by functional category, and second, by focusing on factors previously identified in a meta-analysis of CWP [15]. The second stage employed factor analysis to investigate relationships among inflammatory factors in the Olink panel, followed by the use of factor scores derived from this analysis to determine whether these composite measures predict CWP. In the third stage, we examined the genetic and environmental covariance structure among a subset of inflammatory factors, leveraging the twin pair relationships within the data. Finally, in the fourth stage, we sought to elucidate the underlying network of relationships among this subset of inflammatory factors.

### 2.5. Statistical Analysis

Association analyses of CWP with each of the 70 inflammatory factors, as well as CRP, ApoA1, and ApoB, were performed using version 1.1.9 of the glmmTMB package [30] on an R 4.4.1 [31] installation. Factor analysis of Olink inflammatory markers was performed by first imputing missing data using the missForest package (version 1.5) in R [32], with default settings. Residuals were created using linear regression to remove the effects of age and age-squared. Principal factor analysis of inflammatory factors was then conducted on these residual Olink scores, followed by Promax rotation, using the fa function in the R package psych, version 2.4.6.26 [33]. Factor scores derived from this analysis were then used in linear mixed-effects models again with the glmmTMB package (version 1.1.10) [30].

Twin modelling was conducted to estimate genetic and environmental (shared and nonshared among twin pairs) variance and covariance, using OpenMx (version 2.21.13) [34,35] and the umxACE and related functions in the umx R package (version 4.21.0) [36]. These methods employ full-information maximum-likelihood estimation, allowing for data missing at random and without reducing power or introducing bias by excluding incomplete cases.

To investigate the potential causal relationships among the variables studied using a hypothesis-free approach, we employed additive Bayesian network (ABN) models [37] using the R package abn, version 3.1.1 [38], along with the rjags package, version 4.16 [39], to conduct a parametric bootstrap. We followed a four-stage analysis process to develop a robust causal model and minimize overfitting, as outlined in previous work [40]. Briefly, we determined the maximum number of arcs (causal connections) needed for each dependent variable uncovered by the search process by progressively increasing the maximum number of arcs and stopping when the likelihood of the model stops improving. Overfitting was controlled by bootstrapping 100 samples and retaining only arcs present in at least 50% of simulations.

To better approximate normality, CRP and BMI were log-transformed prior to analysis.

## 3. Results

### 3.1. Descriptive Statistics

Descriptive statistics for age, BMI, CRP, ApoA1, ApoB, and the 70 Olink factors, categorized by CWP status, are presented in Appendix A, with the factors grouped into four functional categories. *T*-tests for BMI, several pro-inflammatory factors (CCL23, CCL3, IL.18R1, TNFB, and TWEAK), anti-inflammatory factors (ApoA1 and IL10), immunoregulatory factors (ADA and CCL19), and multifunction factor CDCP1 are nominally significant (*p* < 0.05); however, it is important to note that these tests do not account for within-family correlation, as the data are from twin pairs. Additionally, a multiple testing correction should be applied.

### 3.2. Mixed-Effects Logistic Regression of Individual Inflammatory Factors on CWP

Mixed-effects logistic regression models were used to regress CWP separately on each of the 70 Olink inflammatory factors, as well as CRP, ApoA1, and ApoB, with age and age-squared included as covariates. To assess whether any signal could be detected in the regression of CWP on the 73 inflammatory factors, QQ plots were created to compare the observed (unadjusted) *p*-values with those expected under the null hypothesis. Each of the functional categories was examined separately (Figure 1). The individual regression coefficients, along with standard errors, test statistics, and *p*-values, are presented in Table 1. Three variables appeared nominally significant: the pro-inflammatory factor IL.18R, the anti-inflammatory factor ApoA1, and the multifunctional factor CDCP1 (in bold). However, after applying false discovery rate corrections within each group, none of these factors remained significant at *p* < 0.05.

Given the prior association of BMI with several inflammatory factors, as well as with CWP, additional stepwise analyses were conducted (see Appendix A). Regression models for CWP were run first without covariates (except for family), then with age, age-squared, and finally log(BMI), as these covariates are associated with CWP and/or inflammatory factors. Including BMI in the model reduced the effect size (and increased the *p*-value) for IL.18R and ApoA1, but not for CDCP1. BMI always had a large effect and was highly statistically significant.

Since the analysis of all 73 inflammatory factors did not provide evidence for associations with CWP after multiple testing corrections, we next focused on the six inflammatory factors identified in a recent meta-analysis [15] (Table 2), which had prior evidence of differences between cases and controls. However, none was statistically significant, with or without covariates in the model, although CRP was suggestively significant (*p* < 0.08); however, this effect disappeared when including BMI in the model. Table 2 also contains area under the curve (AUC) and McFadden’s adjusted R^2^, illustrating that prediction was not very good, but including BMI in the model improves prediction substantially.

### 3.3. Factor Analysis

Factor analyses were performed on the 70 Olink inflammatory biomarkers measured in 2346 individuals to reduce the complexity of the data and increase statistical power for association analyses with CWP. Based on inspection of a scree plot of eigenvalues (see Appendix A), seven factors were deemed sufficient to summarize the data, although 14 factors would be required if the criterion of eigenvalues greater than one were applied. The factor loadings for the seven factors extracted are presented in Appendix A. CWP was regressed on the factor scores derived from these seven factors, controlling for age at testing, age-squared, log(BMI), and the random effect of family, using a mixed-effects model. No evidence of an association was found between CWP and any of the inflammatory marker factor scores, either in separate regressions or in a combined model with all seven factors as predictors, after controlling for the large effect of BMI (Appendix A).

### 3.4. Twin Modelling

Figure 2 presents the estimated phenotypic correlation matrix among 10 variables: CWP, BMI, ApoA and ApoB1, and the six inflammatory factors from the meta-analysis listed in Table 2. The most prominent block of correlations is among IL6, TNF, IL8, IL10, and IFN_gamma, the Olink factors in the model. Additionally, the correlation between CRP and BMI is substantial at 0.41. The largest correlation involving CWP is with BMI, at 0.23.

Twin models partition phenotypic covariance (or correlations when presenting standardized results) into genetic, shared environmental, and nonshared environmental covariance matrices, which sum to the phenotypic correlation matrix when standardized. Figure 3 contains the phenotypically standardized genetic covariances in the upper triangle, genetic correlations in the lower triangle, and estimated heritabilities along the diagonal. Briefly, heritability is the proportion of variance in a trait explained by genetic differences between people across the genome, and genetic covariances represent the extent to which that genetic variance is shared by a pair of traits. An overall likelihood ratio test found that genetic variance and covariance cannot be dropped from the model without a significant degradation in fit (*χ*^2^_55_ = 92.6, *p* < 0.001). The most heritable phenotypes were BMI at 0.72 and CRP at 0.50. The inflammatory factors had low (IL6, 0.06) to moderate (ApoA1 and B, 0.42) heritabilities. The most notable genetic covariances are between CWP and BMI, at 0.16, which therefore explains the vast majority of the 0.23 phenotypic correlation. The genetic covariance between BMI and CRP is 0.29, explaining three-quarters of the phenotypic correlation. The correlations among the Olink factors did not appear to be substantially explained by genetic covariance. While some genetic correlations involving Olink variables were notable (lower triangle of the matrix), these should be interpreted cautiously, particularly when based on low heritabilities, as they may not carry substantial meaning.

Figure 4 contains the phenotypically standardized shared environmental covariances in the upper triangle, shared environmental correlations in the lower triangle, and estimated shared environmental variances along the diagonal. Like genetic variance, these variances and covariances represent the proportion of phenotype variance and covariance explained by twin pairs sharing environments (e.g., being reared together and potentially having common environmental influences in adulthood). Overall, this component of covariance was not statistically significant (*χ*^2^_55_ = 16.9, *p* = 1). However, examining individual components, the variable with the most shared environmental variance was CWP at 0.30. Nonetheless, a test of shared environmental variance of CWP and overall covariance with other measures indicated that it was not significant (*χ*^2^_10_ = 8.95, *p* > 0.5). None of the strong phenotypic correlations is explained by shared environmental covariance. In addition, in some cases, the shared environmental covariance is in the opposite direction to that of the genetic covariance, effectively cancelling out some of the contribution to the phenotypic correlation. Furthermore, the substantial shared environmental correlations should be interpreted with caution, given the small and statistically nonsignificant shared environmental variance components.

Figure 5 contains the phenotypically standardized nonshared environmental covariances in the upper triangle, nonshared environmental correlations in the lower triangle, and estimated nonshared environmental variances along the diagonal. This component of variance represents the effect of environmental factors that are unique to individual members of a twin pair. A test of all the nonshared environmental covariances as a set found them to be necessary in the model (*χ*^2^_45_ = 190.9, *p* < 0.001). While most nonshared environmental covariances were small, the majority of the variance in the Olink factors is nonshared environmental, and therefore, the correlations among those Olink factors appear predominantly due to nonshared environmental covariance, with these covariances being statistically significant as a set (*χ*^2^_35_ = 165.1, *p* < 0.001).

### 3.5. Causal Modelling

We first fitted an ABN model to log(CRP), log(BMI), and the six inflammatory markers from the meta-analysis. In this and the following model, we forced the direction of causation to be from BMI to CWP, rather than the reverse, given that this is the more plausible causal direction. It should be noted that for any model that is uncovered by ABN, there is a corresponding model with causal arrows in the opposite direction with an equivalent fit. For this initial model, a maximum of three causal variables were required for each dependent variable. A directed acyclic graph (DAG) of the resulting model, following bootstrapping, is shown in Figure 6. CWP is found to be directly linked only to BMI, which in turn is linked to CRP, which in turn is linked to IL6, which is linked to IFN.gamma, which is connected to a network of the remaining three inflammatory markers.

Next, fitting an ABN model to log(CRP), log(BMI), ApoA1, ApoB, and CWP (the three non-Olink inflammatory variables) resulted in only a single causal variable for each dependent variable needed. A directed acyclic graph depicting the resulting model following bootstrapping is shown in Figure 7. CWP was only linked to BMI, which in turn was linked to ApoA1. CRP and ApoB were also linked.

## 4. Discussion

The largest sample to date reporting on the association between proteomics and CWP has found no evidence for association, after adjusting for multiple testing. Considering only factors with some prior evidence for association still resulted in no significant association with any inflammatory factor. Since IL-10 was not found to be significantly associated with fibromyalgia in the meta-analysis, it is unsurprising that no association is found in the present sample. However, the other five factors examined had varying degrees of prior evidence of association [15]. The total meta-analysis sample for IL-6 was of similar size to the present TwinsUK sample, yet we found no evidence of association. This is also true for TNF and IL-8. The meta-analysis total sample for IFNgamma was much smaller than the TwinsUK sample, yet the TwinsUK sample again shows no evidence of association, in contrast to the significant *p*-value obtained in the meta-analysis. The meta-analysis of CRP involved a much larger total sample than TwinsUK, and the present study finds only suggestive evidence of association, with CRP levels being higher in CWP cases, but the test of association is not quite significant (*p* < 0.08). CRP may be truly associated with CWP, but our sample might be of insufficient size to detect the small effect. Additionally, the relationship completely disappears after controlling for BMI.

These findings contribute to the evolving understanding of the complex interplay between chronic pain conditions, specifically CWP, and systemic inflammation. The absence of significant association of inflammatory factors in the largest study to date suggests that the aetiology of CWP may involve intricate mechanisms beyond the scope of these inflammatory factors. It is crucial to recognize that CWP represents a multifaceted condition, encompassing not only physiological but also psychological and societal dimensions. The lack of a stronger inflammatory signature challenges the simplistic notion of CWP as an inflammatory disorder, emphasizing the need for a holistic approach in deciphering its underlying mechanisms.

The lack of evidence linking inflammatory factors with CWP may not be entirely surprising, as there is evidence suggesting the relationship between inflammation and CWP may be mediated primarily by BMI. A meta-analysis supports the robust association between BMI and CRP, with a correlation 0.36 [41]. Notably, these associations were more pronounced in women, emphasizing the gender-specific influence of obesity on CRP levels, and we found a correlation of 0.41 in the present all-female sample. Apolipoproteins are also implicated in obesity progression through mechanisms involving lipid metabolism, energy expenditure, and inflammatory responses. Furthermore, there is evidence that CRP plays a causal role in obesity. CRP not only serves as a marker of inflammation but may play an active role in metabolic dysregulation. Elevated CRP levels are linked with altered apolipoprotein concentrations, as demonstrated by experimental studies in transgenic rats. Specifically, apolipoproteins such as ApoA1, ApoA2, ApoA4, ApoC1, ApoC2, ApoC4, ApoE, and ApoM were significantly increased, while ApoB-100 levels were reduced in these rats, alongside heightened triglyceride levels, compared to controls [42]. In addition, CRP regulates the expression of key adipokines and inflammatory mediators, including adiponectin, TNF-α, leptin, IL-6, and PPAR-γ, which may underpin its involvement in insulin resistance, obesity, and metabolic syndrome. These factors may, therefore, be upstream of BMI, with BMI being the primary cause of CWP.

The ABN models support this hypothesis. CWP appears to only be directly linked to BMI in the models uncovered, with BMI linked to inflammatory factors. It is important to note that for any ABN model that is uncovered, there is an alternative model that would fit equally well but with causal paths reversed. More specifically, the method does not shed light on whether BMI causes CWP, or the reverse, or if causation is bidirectional. What the model does show is that the two variables are directly associated. The method is best to be considered as an approach to elucidate networks connecting variables but not causal direction. The ABN results should be considered when examining the twin modelling, which uncovered significant heritability for CWP as well as the inflammatory factors examined and significant genetic covariance among CWP and several inflammatory factors. The ABN results suggest that these covariances of CWP with inflammatory factors are all mediated by BMI, primarily due to genetic influences but also nonshared environmental covariances.

It is important to acknowledge the complexity and heterogeneity of CWP. Our blood samples were obtained at a single point in time, not necessarily correlated with individuals’ expression of CWP severity. Perhaps if symptom severity can be tracked and blood samples obtained at multiple intervals, differences could be detected, particularly if the extent of inflammation varies with the level of pain at a particular time. Future research should explore additional molecular and genetic factors as well as environmental influences that may contribute to the variability in CWP presentations. For instance, the potential role of non-inflammatory biomarkers, such as neurological or metabolic factors, in the aetiology of CWP remains to be examined. Diet and exercise have an impact on BMI and may have both direct and indirect effects on inflammation and pain. By examining all these factors in a longitudinal context, the underlying aetiology of CWP could be better understood, and causal inferences could potentially be made, in addition to merely detecting associations. The absence of significant findings suggests the need for a more thorough examination of clinical factors and perhaps the presence of other important covariates masking any relationship with inflammation that may be present.

## 5. Conclusions

Our findings do not provide strong evidence directly linking CWP with inflammatory markers in the present sample. However, the observed associations reinforce the relationship between BMI and CWP, as well as the well-established link between increased BMI and systemic inflammation. This suggests that BMI or adiposity may act as a central mediator in the relationship between inflammation and CWP, highlighting the need for future research to account for its potential role in both pathways. Given the complex and multifactorial nature of CWP, comprehensive investigations are essential. Large-scale, longitudinal studies incorporating a broad range of potential mediators and covariates will be crucial to disentangle these mechanisms. Furthermore, multicentre collaborations are needed to validate these findings across diverse populations, enhance statistical power, and explore alternative biological and environmental pathways contributing to CWP. Clinically, these findings underscore the importance of considering BMI in the assessment and management of CWP. Targeting excess adiposity through weight management and lifestyle interventions may be a viable strategy to mitigate inflammation and potentially alleviate CWP symptoms, warranting further exploration in clinical settings.

## Figures and Tables

**Figure 1 biomolecules-15-00155-f001:**
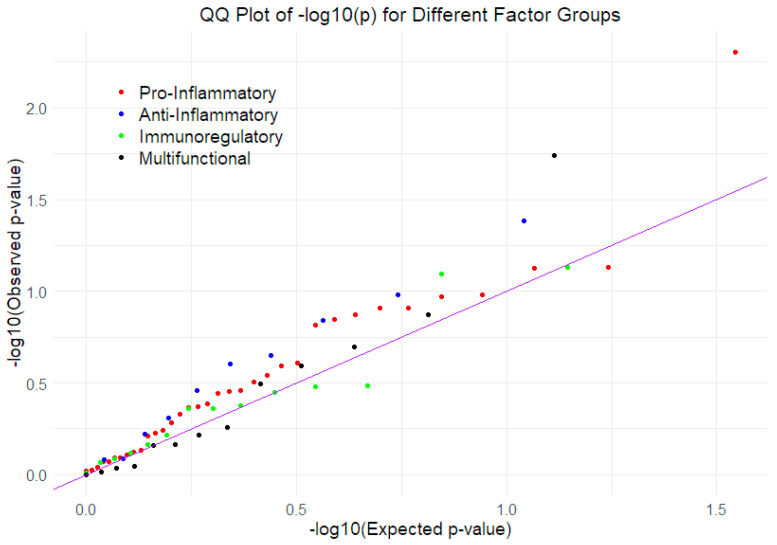
Four QQ plots of *p*-values for CWP regressed on 73 inflammatory factors, grouped by function (pro-inflammatory, anti-inflammatory, immunoregulatory, and multifunctional), controlling for age and age-squared, as well as family membership. This demonstrates lack of signal detected when considering multiple testing of four sets of inflammatory factors, with all observed *p*-values not differing much from that expected under the null hypothesis.

**Figure 2 biomolecules-15-00155-f002:**
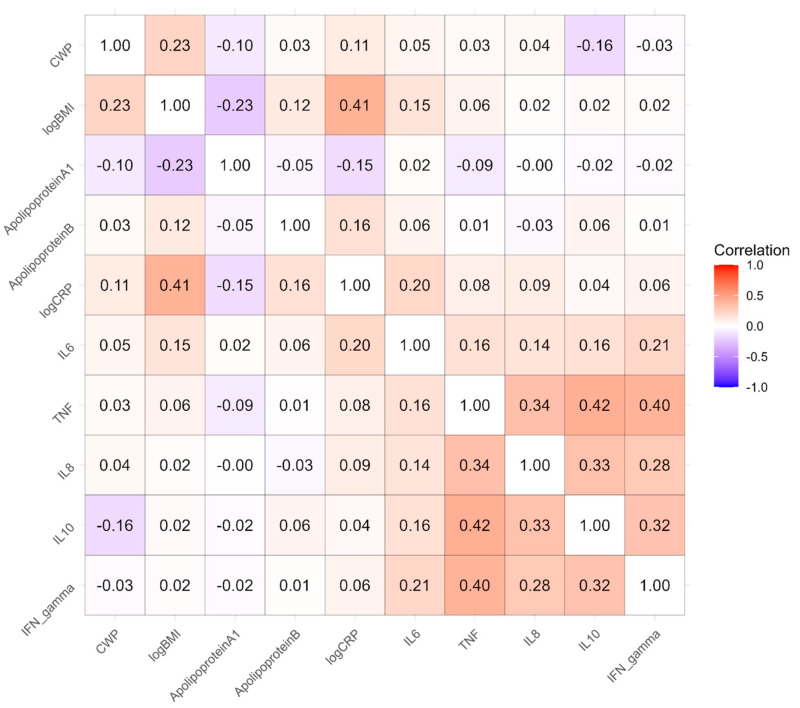
Phenotypic correlation matrix among 10 variables obtained from estimating genetic, shared environmental, and nonshared environmental covariances in a multivariate twin model.

**Figure 3 biomolecules-15-00155-f003:**
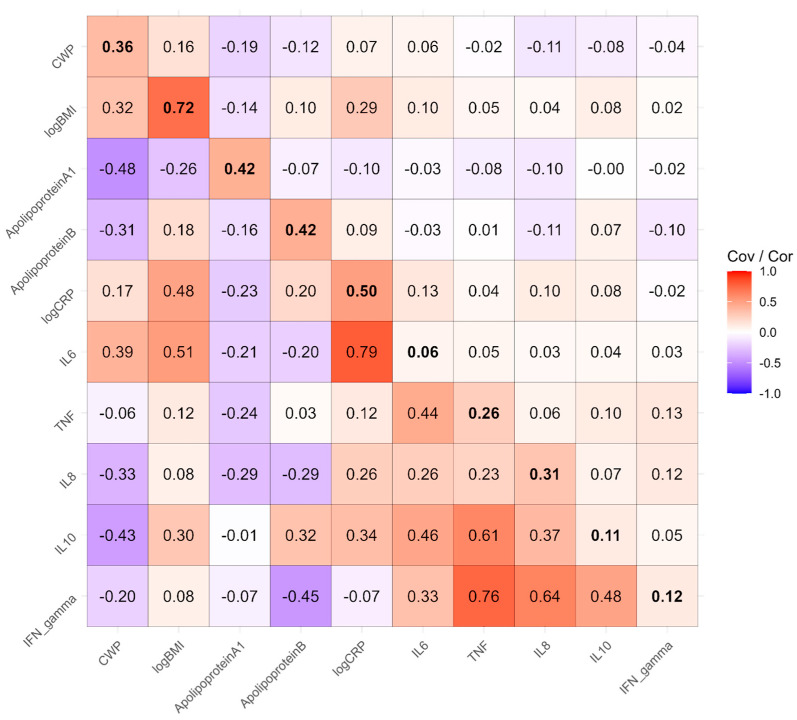
Phenotypically standardized genetic covariance matrix in the upper triangle, genetic correlation matrix in the lower triangle, with heritabilities along the diagonal in bold.

**Figure 4 biomolecules-15-00155-f004:**
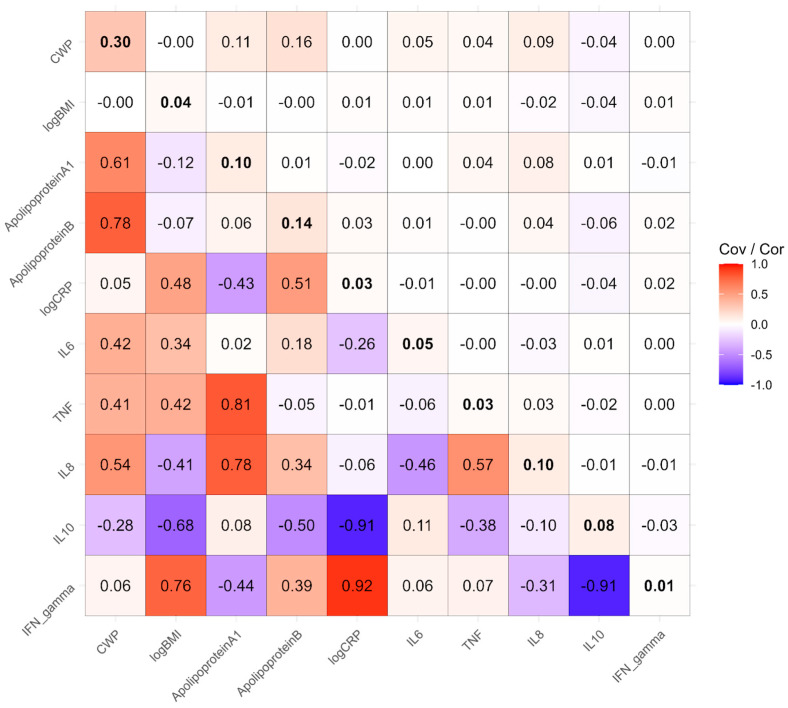
Phenotypically standardized shared environmental covariance matrix in the upper triangle, shared environmental correlation matrix in the lower triangle, with shared environmental variances along the diagonal in bold.

**Figure 5 biomolecules-15-00155-f005:**
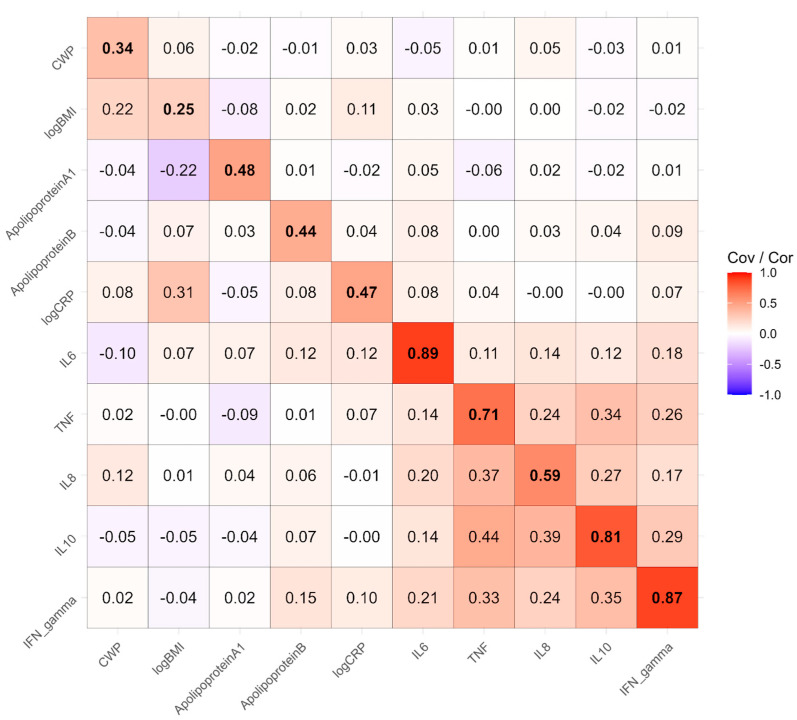
Phenotypically standardized nonshared environmental covariance matrix in the upper triangle, nonshared environmental correlation matrix in the lower triangle, with nonshared environmental variances along the diagonal in bold.

**Figure 6 biomolecules-15-00155-f006:**
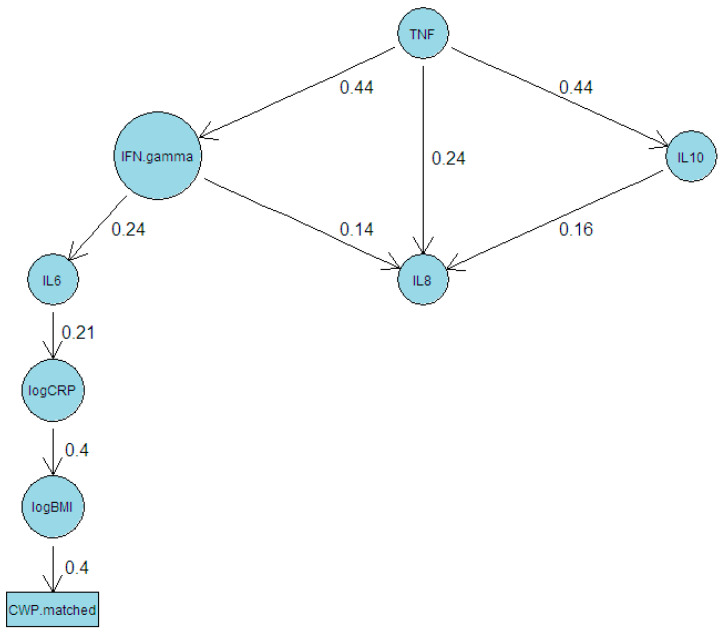
Directed acyclic graph resulting from fitting an additive Bayesian network model to six inflammatory factors plus log(BMI) and log(CWP).

**Figure 7 biomolecules-15-00155-f007:**
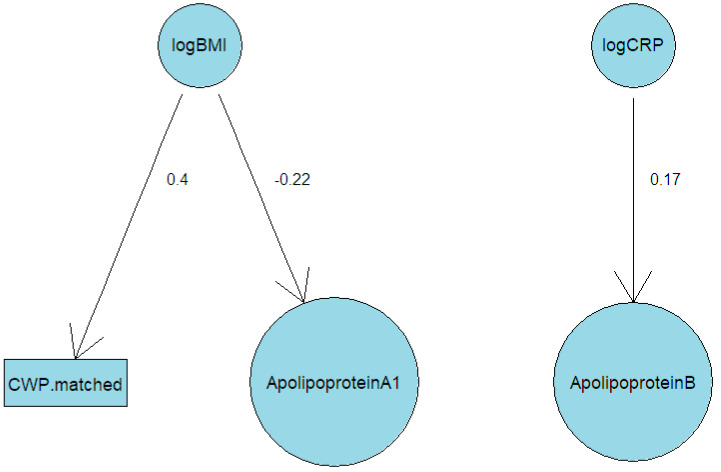
Directed acyclic graph resulting from fitting an additive Bayesian network model to log(CRP), log(BMI), ApoA1, ApoB, and CWP.

**Table 1 biomolecules-15-00155-t001:** Mixed-model regression of CWP on each of the 73 inflammatory factors, controlling for age, age-squared, and family membership, grouped by four categories of inflammatory factors: (1) pro-inflammatory molecules, which are essential for initiating the immune response to infectious agents; (2) anti-inflammatory molecules, which are important in limiting inflammation and preventing tissue damage; (3) immunoregulatory molecules, which possess both pro- and anti-inflammatory activities; and (4) multifactorial molecules, which are defined as those performing various functions. See text for further information. *p*-values that are nominally significant at the 0.05 level are highlighted in bold.

Factor	N	Estimate	SE	z	*p*	Adj p
**Pro-inflammatory factors**						
log(CRP)	870	0.432	0.242	1.785	0.074	0.535
ApolipoproteinB	716	−0.221	0.278	−0.794	0.427	0.757
CCL11	904	0.061	0.251	0.244	0.807	0.930
CCL23	904	−0.365	0.249	−1.467	0.142	0.535
CCL3	896	0.400	0.246	1.623	0.105	0.535
CCL4	895	0.276	0.238	1.160	0.246	0.739
CD40	904	0.127	0.253	0.502	0.616	0.862
CXCL1	904	−0.067	0.245	−0.274	0.784	0.930
CXCL10	904	0.159	0.246	0.646	0.518	0.824
CXCL11	902	−0.139	0.247	−0.563	0.573	0.862
CXCL5	904	−0.371	0.241	−1.540	0.124	0.535
CXCL6	904	0.059	0.249	0.236	0.813	0.930
CXCL9	903	0.382	0.267	1.430	0.153	0.535
DNER	902	−0.048	0.248	−0.195	0.846	0.930
EN.RAGE	904	−0.359	0.239	−1.498	0.134	0.535
IFN.gamma	898	0.183	0.223	0.821	0.412	0.757
IL.12B	904	−0.287	0.253	−1.137	0.255	0.739
IL.15RA	782	0.431	0.242	1.784	0.074	0.535
IL.17C	795	0.238	0.236	1.010	0.312	0.739
IL.18R1	904	0.758	0.270	2.808	**0.005**	0.175
IL18	903	−0.225	0.245	−0.917	0.359	0.739
IL7	879	−0.018	0.254	−0.070	0.944	0.956
IL8	901	0.126	0.238	0.528	0.597	0.862
MCP.1	901	0.223	0.240	0.930	0.353	0.739
MCP.2	902	0.236	0.252	0.940	0.347	0.739
MCP.4	903	−0.078	0.246	−0.316	0.752	0.930
MMP.1	904	−0.266	0.250	−1.067	0.286	0.739
SLAMF1	830	0.388	0.252	1.542	0.123	0.535
TNF	901	0.012	0.216	0.055	0.956	0.956
TNFB	900	−0.028	0.252	−0.112	0.911	0.956
TNFSF14	903	0.187	0.238	0.784	0.433	0.757
TRAIL	904	0.405	0.251	1.613	0.107	0.535
TWEAK	904	−0.046	0.243	−0.188	0.851	0.930
uPA	904	−0.083	0.249	−0.332	0.740	0.930
VEGFA	903	0.177	0.243	0.728	0.466	0.777
**Anti-inflammatory factors**						
ApolipoproteinA1	716	−0.612	0.300	−2.041	**0.041**	0.454
FGF.21	897	−0.052	0.243	−0.213	0.831	0.914
HGF	904	0.286	0.249	1.150	0.250	0.551
IL.10RB	902	0.412	0.254	1.622	0.105	0.530
IL10	896	−0.284	0.233	−1.219	0.223	0.551
LAP.TGF.beta.1	902	−0.055	0.241	−0.227	0.821	0.914
LIF.R	902	0.361	0.248	1.459	0.145	0.530
MMP.10	904	−0.232	0.247	−0.941	0.347	0.636
SIRT2	821	−0.199	0.287	−0.693	0.489	0.768
STAMBP	904	−0.137	0.263	−0.519	0.604	0.830
TRANCE	904	−0.006	0.244	−0.026	0.979	0.979
**Immunoregulatory factors**						
ADA	904	−0.007	0.253	−0.027	0.978	0.978
CCL19	904	0.374	0.214	1.748	0.080	0.563
CCL20	899	−0.237	0.242	−0.977	0.329	0.764
CCL25	904	0.214	0.265	0.807	0.419	0.764
CCL28	900	−0.075	0.249	−0.300	0.764	0.925
CD244	904	0.233	0.252	0.926	0.354	0.764
CD5	904	0.057	0.250	0.226	0.821	0.925
CD6	904	0.236	0.243	0.973	0.330	0.764
CD8A	904	−0.097	0.236	−0.409	0.683	0.925
CX3CL1	903	0.446	0.249	1.789	0.074	0.563
Flt3L	903	0.128	0.249	0.515	0.607	0.925
IL6	580	−0.283	0.364	−0.778	0.436	0.764
PD.L1	903	−0.187	0.240	−0.780	0.435	0.764
TNFRSF9	902	−0.045	0.255	−0.177	0.859	0.925
**Multifunctional factors**						
AXIN1	904	−0.010	0.262	−0.037	0.971	0.999
CASP.8	901	0.140	0.237	0.594	0.553	0.999
CDCP1	901	0.683	0.289	2.364	**0.018**	0.235
CSF.1	903	0.282	0.248	1.138	0.255	0.829
CST5	901	−0.264	0.265	−0.995	0.319	0.831
FGF.19	904	0.024	0.237	0.102	0.919	0.999
NT.3	898	0.090	0.221	0.409	0.683	0.999
OPG	904	−0.034	0.262	−0.128	0.898	0.999
OSM	904	0.000	0.234	0.002	0.999	0.999
SCF	901	−0.271	0.212	−1.278	0.201	0.829
ST1A1	856	0.135	0.261	0.517	0.605	0.999
TGF.alpha	902	−0.094	0.238	−0.397	0.692	0.999
X4E.BP1	904	−0.350	0.234	−1.495	0.135	0.829

**Table 2 biomolecules-15-00155-t002:** Mixed-model regression of CWP on each inflammatory factor reported in a meta-analysis [15], sequentially controlling for age, age-squared, and log(BMI), as well as family membership. Sample size (N), parameter estimate, standard error (SE), *z*-test, unadjusted *p*-value, area under the curve (AUC), and McFadden’s adjusted R^2^ are presented for each model.

Factor	Covariates	N	Estimate	SE	z	*p*	AUC	adj R^2^
IL6	No covariates	580	−0.282	0.362	−0.779	0.436	0.542	0.002
	Age		−0.285	0.364	−0.783	0.433	0.548	0.002
	Age + Age^2^		−0.283	0.364	−0.778	0.436	0.531	0.002
	Age + Age^2^ + log(BMI)		−0.419	0.393	−1.065	0.287	0.591	0.012
TNF	No covariates	901	0.034	0.210	0.162	0.871	0.500	0.000
	Age		0.013	0.217	0.060	0.952	0.526	0.000
	Age + Age^2^		0.012	0.216	0.055	0.956	0.610	0.002
	Age + Age^2^ + log(BMI)		−0.057	0.224	−0.254	0.799	0.647	0.020
IL8	No covariates	901	0.158	0.233	0.679	0.497	0.518	0.001
	Age		0.146	0.237	0.617	0.537	0.531	0.001
	Age + Age^2^		0.126	0.238	0.528	0.597	0.610	0.003
	Age + Age^2^ + log(BMI)		0.152	0.249	0.607	0.544	0.645	0.020
log(CRP)	No covariates	870	0.435	0.242	1.795	0.073	0.563	0.005
	Age		0.432	0.243	1.777	0.076	0.564	0.005
	Age + Age^2^		0.432	0.242	1.785	0.074	0.613	0.007
	Age + Age^2^ + log(BMI)		0.084	0.278	0.302	0.763	0.650	0.021
IL10	No covariates	896	−0.274	0.233	−1.175	0.240	0.537	0.002
	Age		−0.287	0.235	−1.223	0.221	0.550	0.002
	Age + Age^2^		−0.284	0.233	−1.219	0.223	0.599	0.004
	Age + Age^2^ + log(BMI)		−0.312	0.242	−1.287	0.198	0.658	0.021
IFN.gamma	No covariates	898	0.199	0.220	0.906	0.365	0.511	0.001
	Age		0.189	0.224	0.844	0.399	0.520	0.001
	Age + Age^2^		0.183	0.223	0.821	0.412	0.581	0.003
	Age + Age^2^ + log(BMI)		0.178	0.230	0.774	0.439	0.645	0.019

## Data Availability

The TwinsUK data that support the findings of this study are available by application from https://twinsuk.ac.uk/resources-for-researchers/access-our-data/ (accessed on 24 December 2024). Restrictions apply to the availability of these data, which were used under licence for this study.

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
