# Peer review of "A Genetic and Environmental Analysis of Inflammatory Factors in Chronic Widespread Pain Using the TwinsUK Cohort"

_biomolecules, 2025, doi:10.3390/biom15020155_

Round 1

Reviewer 1 Report

Comments and Suggestions for Authors

In the manuscript entitled “Association of inflammatory factors with chronic widespread pain” the authors aimed to study the contribution of inflammation to CWP.  Prior to acceptance for publication, the following points need to be addressed:

1.    Please, add a table of the characteristics of females included in the study. Age, age range for controls and the CWP group.

2.    Include comparisons of the analytes measured in an age-matched control group of females.

3.    Calculate ROC between the control and CWP groups for each analyte. What is the sensitivity and specificity of the potential biomarkers studied?

4.    What is the adjusted R2 for the model in table 1?

5.    The authors should consider a stepwise regression to identify what analytes are more suited for a model to explain CWP.

6.    Were models chosen based on the AIC, BIC or what?

7.    Why was the factor analysis performed in 2,346 individuals if only females with CWP were considered in this study?

8.    The correlations in figure 2 are weak. It is not clear what is the purpose of this figure? Figures 3 and 4 show better correlations. What was the rationale for choosing those cytokines for the matrix?

9.    Add to the discussion more background on CWP.

10. Add to the discussion more background on each of the significant cytokines studied.

Reviewer 2 Report

Comments and Suggestions for Authors

Thank you for the opportunity to review the manuscript, "Association of inflammatory factors with chronic widespread pain." Below are my comments and suggestions for improving the manuscript.

General Comments

  • The manuscript is thorough and addresses an important topic, but it could benefit from more specific discussions and contextual interpretations of the findings.
  • Including a section on potential clinical implications and future research directions would strengthen the discussion and conclusion.

Title

  • Consider revising the title to better reflect the study design, e.g., "A Genetic and Environmental Analysis of Inflammatory Factors in Chronic Widespread Pain Using the TwinsUK Cohort."

Narrative Structure

1. Missed or Misinterpreted Trends or Patterns

  • Lines 53-62: Expand the discussion on the broader implications of inflammatory factors like CRP and ApoA1 in relation to obesity and CWP. How do these findings compare with other pain conditions?
  • Lines 247-261: Provide a more detailed interpretation of the genetic covariance between BMI, CRP, and CWP. What are the potential biological pathways involved?
  • Lines 343-347: Discuss the role of gender differences in the relationship between BMI and CWP, as highlighted in the literature.

2. Under or Over-Emphasized Results

  • Lines 198-200: Provide more context on why CDCP1 emerged as nominally significant despite adjustments. Could it represent a unique pathway?
  • Lines 333-334: Elaborate on the “suggestive” association of CRP with CWP. What are the implications for future biomarker studies?

3. Organizational Flow Improvements

  • Improve transitions between sections, particularly from Methods to Results and from Results to Discussion.
  • Ensure consistent formatting of subheadings to improve readability.

4. Incorporate Recent Literature

  • Include references that contextualize the role of inflammation in CWP and similar conditions. Examples:
    • PMID: 30128914
    • PMID: 37731935

Description of Methods

1. Sufficiency of Method Explanations

  • Lines 97-102: Provide more details on the PEA technology, particularly its sensitivity and specificity relative to other proteomic methods.
  • Lines 153-162: Clarify how confounders like BMI were modeled statistically. Were interaction effects considered?

Tables and Figures

1. Recommendations for Improvements

  • Figure 1: Enhance the captions by explaining the clinical relevance of the findings in the QQ plots.
  • Table 2: Include additional descriptive data on patient demographics (e.g., BMI range, comorbidities) to provide context for the findings.

2. Titles, Descriptions, or Labels

  • Ensure all figures and tables have comprehensive captions. For example:
    • Table 1: Include a brief explanation of the functional categories of inflammatory factors.
    • Figure 3: Label the axes clearly and explain the significance of genetic covariances.

Discussion

1. Limitations

  • Lines 375-377: Expand on the limitations related to the cross-sectional design and the single-time-point blood samples. How might these limitations affect the conclusions?
  • Discuss the potential confounding effects of unmeasured variables like diet or physical activity.

2. Suggestions for Future Research

  • Investigate the temporal relationship between BMI, CRP, and CWP through longitudinal studies.
  • Explore the potential role of non-inflammatory biomarkers, such as neurological or metabolic factors, in the etiology of CWP.

Conclusion

  • Reiterate the importance of considering BMI as a central mediator in the relationship between inflammation and CWP.
  • Highlight the need for multicenter studies to validate the findings and explore alternative pathways.

Round 2

Reviewer 1 Report

Comments and Suggestions for Authors

My previous concerns have been addressed.